# Reducing Lung ATP Levels and Alleviating Asthmatic Airway Inflammation through Adeno-Associated Viral Vector-Mediated CD39 Expression

**DOI:** 10.3390/biomedicines9060656

**Published:** 2021-06-08

**Authors:** Yung-An Huang, Jeng-Chang Chen, Chih-Ching Wu, Chia-Wei Hsu, Albert Min-Shan Ko, Li-Chen Chen, Ming-Ling Kuo

**Affiliations:** 1Department of Microbiology and Immunology, Graduate Institute of Biomedical Sciences, College of Medicine, Chang Gung University, Taoyuan 33302, Taiwan; D0301203@cgu.edu.tw; 2Department of Medicine, University of California, San Diego, CA 92093, USA; 3Department of Surgery, Chang Gung Memorial Hospital-Linkou, College of Medicine, Chang Gung University, Taoyuan 33302, Taiwan; bx9619@cgmh.org.tw; 4Department of Medical Biotechnology and Laboratory Science, College of Medicine, Chang Gung University, Taoyuan 33302, Taiwan; luckywu@mail.cgu.edu.tw; 5Molecular Medicine Research Center, Chang Gung University, Taoyuan 33302, Taiwan; 6Department of Otolaryngology—Head and Neck Surgery, Chang Gung Memorial Hospital-Linkou, Taoyuan 33302, Taiwan; allenshu828@gmail.com; 7Agricultural Biotechnology Research Center, Academia Sinica, Taipei 11574, Taiwan; 8Department of Cardiovascular Diseases, Chang Gung Memorial Hospital-Linkou, Taoyuan 33302, Taiwan; koalbert@hotmail.com; 9Department of Pediatrics, Division of Allergy, Asthma, and Rheumatology, Chang Gung Memorial Hospital-Linkou, Taoyuan 33302, Taiwan; lcchen@adm.cgmh.org.tw; 10Department of Pediatrics, New Taipei Municipal TuCheng Hospital, New Taipei City 23664, Taiwan

**Keywords:** apyrase, adeno-associated viral vector, airway hyperresponsiveness, cytokine, regulatory T cells

## Abstract

Asthma is a chronic respiratory inflammatory disease. Patients usually suffer long-term symptoms and high medical expenses. Extracellular ATP (eATP) has been identified as a danger signal in innate immunity and serves as a potent inflammatory mediator for asthma. Hydrolyzing eATP in lungs might be a potential approach to alleviate asthmatic inflammation. Recombinant adeno-associated virus (rAAV) vectors that contain tissue-specific cap protein have been demonstrated to efficiently transfer exogenous genes into the lung tissues. To test anti-inflammation efficacy of rAAV-mediated CD39 gene transfer, rAAV-CD39 was generated and applied to OVA-mediated asthmatic mice. BALB/c mice were sensitized intraperitoneally and challenged intratracheally with OVA and treated with rAAV-CD39. At the end of procedure, some inflammatory features were examined. rAAV-CD39 treatment downregulated the levels of pulmonary eATP by the rescued expression of CD39. Several asthmatic features, such as airway hyperresponsiveness, eosinophilia, mucin deposition, and IL-5/IL-13 production in the lungs were decreased in the rAAV-CD39-treated mice. Reduced IL-5/IL-13 production and increased frequency of CD4^+^FoxP3^+^ regulatory T cells were detected in draining lymph nodes of rAAV-CD39 treated mice. This evidence suggested that rAAV-mediated CD39 gene transfer attenuated the asthmatic airway inflammation locally. The results suggest that rAAV-CD39 might have therapeutic potential for asthma.

## 1. Introduction

Asthma affects more than 300 million people worldwide [1]. Asthma is considered as a chronic inflammatory disease with airway obstruction, excessive mucus production, pulmonary eosinophilia, and airway hyperresponsiveness (AHR) caused by variable allergens.

Although the importance of Th2 responses in adaptive responses has been well documented in the asthma pathogenesis [2], the involvement of endogenous damage-associate molecule patterns (DAMPs), such as uric acid [3], HMGB1 [4], IL-33 [5], or extracellular ATP (eATP) [6,7] in allergic diseases has been described. Among these DAMPs, eATP has been implicated as a critical mediator in several pulmonary diseases, including asthma [6], pulmonary fibrosis [8], and chronic obstructive pulmonary disease [9]. In steady state, the level of eATP is tightly controlled by the ectonucleoside triphosphate diphosphohydrolases (NTPDases), which liberates two phosphates to form AMP. The most potent NTPDase is NTPDase-1 (also known as CD39, gene symbol: *entpd1*) [10]. When CD39 works together with CD73 (ecto-5’-nucleotidase), they together convert the danger signal eATP into adenosine [10]. CD39 can be detected on a wide range of cells, including structural cells and immune cells, as well as on regulatory T cells (Tregs) [11].

As CD39 serves a critical immunoregulatory molecule, its expression is usually suppressed upon the presence of inflammation [12,13]. Reduction of CD39 expression in asthmatic or inflammation has been reported [14]. Thus, the exogenous expression of CD39 in lung tissues might lead to a therapeutic outcome for asthma.

Adeno-associated virus (AAV)-based vectors can efficiently transduce exogenous genes and allow their long-term expression. AAV is a small single-stranded DNA virus characterized by its non-integrating episomal expression and lack of pathogenicity to humans [15]. The tissue tropism of AAV is also beneficial for targeting specific type of the cells. The capsid of serotype 9 shows a tropism to lung epithelial cells [15]. Although AAV-based gene transfer approaches in modulating cystic fibrosis or asthma have been reported in experimental models [16,17,18,19,20], this vector has not been shown in controlling DAMPs.

Based on the lung tropism and high gene transfer efficiency, we attempted to apply a pseudotyped recombined AAV (rAAV) vector carrying CD39 (rAAV-CD39) to treat the OVA-induced asthmatic mice. The results demonstrate that rAAV-CD39 rescued CD39 gene expression and exhibited a positive therapeutic potential. The effects were mainly shown in lung tissues or draining lymph nodes, but less impact was detected in systemic immune responses. Further, the efficacy of exogenous CD39 on the asthmatic responses might be due to greater Treg frequency detected in lung draining lymph nodes. These data provided evidence for AAV gene therapeutic potential on allergic diseases for targeting innate immune response.

## 2. Materials and Methods

### 2.1. Mice

Female BALB/c mice (8–10 weeks old) were used. Mice were purchased from National Laboratory Animal Center (Taipei, Taiwan) and maintained in the Animal Center of Chang Gung University (Taoyuan, Taiwan). Animal caring and handling were approved by Chang Gung University Institutional Animal Care and Use Committee (CGU15 204, 3 August 2016) and also followed the Guide for the Care and Use of Laboratory Animals.

### 2.2. Generation of Recombinant Adeno-Associated Viral (rAAV) Vectors

The coding sequence of CD39 was amplified by specific primer (Appendix A) and cloned into the multiple cloning site of the pAAV-IRES-hrGFP plasmid (Stratagene, Agilent Technologies, La Jolla, CA, USA). The hybrid rAAV2/9 vectors that can efficiently target lung tissues are composed with AAV2 rep and AAV9 cap genes. The generation of rAAV-CD39 or control vectors was conducted as previously described [19,20].

### 2.3. OVA-Induced Asthmatic Airway Inflammation and Viral Vector Treatment

The mice were sensitized via intraperitoneal injection (IP) of OVA (100 μg; MilliporeSigma, Burlington, MA, USA) with alum (2 mg; Thermo Fisher Scientific, Waltham, MA, USA) and challenged intratracheally (IT) without or with OVA (10 µg in 50 µL normal saline) 5 times in 2 weeks (Figure 1A). In the experiments with virus treatment, virus treated groups received rAAV-CD39 or an equivalent vehicle control vector virus (5 × 10^11^ genome copies) IT on day 16. Most assays were conducted based on 4 repetitive experiments.

### 2.4. Measurement of Airway Hyperresponsiveness (AHR)

Two pulmonary function measuring devices (Buxco, Data Sciences International, St. Paul, MN, USA), Whole Body Plethysmography (WBP) and Resistance and Compliance (RC) were applied to determine the Penh or RI values of OVA-sensitized mice, respectively. AHR was induced via methacholine (Mch; MilliporeSigma) aerosol (0, 10, 20, 30, and 40 mg/mL for WBP; 0, 1.7, 5, 15, and 45 mg/mL for RC) on day 34. The measurement procedure was previously described [19]. The Penh or RI values were performed in separate experiments.

### 2.5. Bronchoalveolar Lavage Fluid (BALF) Collection and Leukocyte Counting

After mice were sacrificed on day 34, the tracheas were cannulated, and the lungs were lavaged with 1 + 2 mL Ca^2+^- and Mg^2+^-free PBS supplemented with 0.1 mM EDTA. The leukocytes in BALF were stained and counted as previously described [19].

### 2.6. ATP Measurement

ATP level in BALF was measured by an ATP determination kit (ATPlite, Perkin Elmer, Waltham, MA, USA). Luminescence signal was determined by a luminometer (Promega, Madison, WI, USA).

### 2.7. Lung Histology

Some lungs without BALF collection were fixed with formaldehyde and stained (H&E and PAS) as previously described [19,20]. Each slide represented the lung tissues from a mouse, two fields of a slide were selected, and then five areas with 100 µm square size around the bronchiole in selected field were analyzed by ImageJ software (V1.51, NIH, USA) with Threshold_Colour plugin. The quantitation of histological changes and mucin deposition were determined by HSB assay (Hue: 180–255, Saturation: 62–255, Brightness: 0–255 for H&E; Hue: 210–215, Saturation: 80–255, Brightness: 0–255 for PAS). The mean intensity of each area was recorded as histological score.

### 2.8. Lymphocyte Cultures and Determination of Cytokines

Splenocytes or MLN cells were stimulated with OVA (100 µg/mL; MilliporeSigma) in RPMI 1640 (Thermo Fisher Scientific) for 6 days. Cytokine levels in BALF or culture supernatants were determined by ProcartaPlex (Thermo Fisher Scientific) or ELISA kits (R&D Systems, Bio-Techne, Minneapolis, MN, USA), respectively.

### 2.9. mRNA Expression

Lung tissues were collected and homogenized. Expression of each gene in the lung tissues was analyzed with quantitative RT-PCR based on the specific primers (Appendix A). The Ct value of the sample genes (*entpd1*, *ccl11*, *ccl24*) were normalized to housekeeping gene (*actb*). The expression levels were presented as a fold change compared to the control group (N).

### 2.10. Flow Cytometry Analysis

MLNs were collected and the cells were suspended. About 2 million MLN cells were stained with fluorescent antibodies (Appendix A). The fluorescent signal was analyzed by Attune NxT Flow Cytometer (Thermo Fisher Scientific).

### 2.11. Statistical Analysis

If not stated otherwise for most experiments, the statistical significance of differences between groups was determined by the Kruskal–Wallis test followed with Dunn’s post hoc test, using Prism software (V9.0.0, Graphpad Software, La Jolla, CA, USA). *p* < 0.05 is considered statistically significant.

## 3. Results

### 3.1. Decreased Level of Extracellular ATP in Lungs by the Rescue of CD39 Expression

As extracellular ATP is a danger signal which amplifies the inflammatory response, it plays a critical role in asthmatic airway inflammation. The level of extracellular ATP is controlled by the ectoenzymes, mainly CD39 (gene symbol: *entpd1*). A significantly high ATP level was induced with OVA allergen (Figure 1A,C), in concert with the downregulated expression of CD39 during inflammation (Figure 1B).

To establish a gene therapy system by targeting this axis, a pseudotyped recombinant adeno-associated viral (rAAV) vector carrying the coding sequence of CD39 (rAAV-CD39) was generated and applied into OVA-induced asthmatic mice (Figure 1A). The results show that the rescued CD39 expression in mice with administration of rAAV-CD39 and the ATP levels in bronchoalveolar lavage fluid (BALF) were decreased (Figure 1B,C). To detect adenine nucleosides and nucleotides in BALF with the exogenous CD39, we performed LC-MRM analysis (Appendix A). The patterns of ATP levels in all experimental groups (Appendix A) were similar to the ATPlite analysis. These results indicate that rAAV-mediated CD39 expression reduced elevated eATP in allergic mice.

### 3.2. Asthmatic Airway Inflammation was Alleviated by the Rescued Expression of CD39

To verify the therapeutic potential of rAAV-CD39, the main asthmatic cardinal features, such as airway hyperresponsiveness (AHR) and eosinophilic infiltration, were analyzed after the last OVA challenge. AHR was measured by two devices: Whole Body Plethysmography and Resistance. The rAAV-CD39-treated mice had significantly reduced AHR and airway resistance (Figure 2A,B, respectively). These results suggest that the acute symptom of asthma, dyspnea, can be alleviated by exogenous CD39 gene transfer.

Leukocyte infiltration of the lungs is another important parameter of allergic inflammation. Total and differential cell counts were measured in BALF. Significantly decreased total cell numbers infiltrated in the lungs of rAAV-CD39-treated mice (Figure 2C). The infiltrated eosinophil cell number was also significantly reduced in rAAV-CD39-treated mice (Figure 2D), although the percentages of eosinophils in rAAV-CD39-treated mice were comparable with vector control mice (Figure 2E). The reduction of infiltrated cells in BALF was also associated with the lower expression of VCAM-1 in the treated mice (Appendix A). However, the expression of main chemokines for eosinophils (*ccl11* and *ccl24*) was not affected (Appendix A). Taken together, the administration of rAAV-CD39 modulated eosinophilia by limiting the leukocyte infiltration.

### 3.3. Exogenous CD39 Gene Expression Alleviated Airway Remodeling

Next, we examined the histological changes by rAAV-CD39. Lung tissue slides were treated with H&E or PAS staining to monitor airway thickness and goblet cell metaplasia, respectively. The results show that the airway walls were thickened and more cells were infiltrated in OVA-sensitized and vector control mice (Figure 3A). According to the quantitative results, the inflammatory index was significantly reduced in rAAV-CD39-treated mice (Figure 3C). PAS staining also exhibited elevated mucin deposition in the disease and vector control groups, suggesting increased goblet cell metaplasia (Figure 3B,D). Fewer goblet cell metaplasia was observed in the rAAV-CD39-treated mice. These results are consistent with rAAV-CD39 treatment reducing airway remodeling.

### 3.4. Reduced Th2 Response in the Lungs of rAAV-CD39 Virus-Treated Mice

To investigate the effect of rAAV-CD39 treatment on the immune responses in lungs, the cytokine levels in BALF were analyzed by ProcartaPlex. Among Th2-type cytokines, IL-5 and IL-13 were significantly reduced in BALF of rAAV-CD39-treated mice (Figure 4B,C), whereas IL-4 was unaffected (Figure 4A). The rAAV-CD39-treated mice had increased levels of IFN-γ, a representative Th1-type cytokine, but this upregulation was only marginally significant (*p* = 0.062) compared with the vector control (Figure 4D). Since extracellular ATP might serve as a secondary signal for NLRP3 inflammasome activation [21], IL-1β and IL-18 were also examined. The concentration of either cytokine was elevated in the allergic mice, but was not affected by rAAV-CD39 treatment (Appendix A). The levels of Th17-type cytokines and epithelial cell-derived cytokines were comparable between all experimental groups (data not shown). The data suggest that elevated CD39 transduced by rAAV vector had potential to modulate Th2 response in the lungs of asthmatic mice.

We next determined whether rAAV-CD39 treatment affected local or system immune responses; allergen-specific Th2 responses were measured from supernatants of OVA-stimulated lymphocyte cultures from draining lymph nodes or spleens. Single-cell suspensions of mediastinal lymph nodes (MLNs) and spleens were cultured with OVA. The levels of Th2-type cytokines in the supernatants were analyzed by ELISA. Similar to BALF, reduced production of IL-5 and IL-13 was observed in OVA-stimulated rAAV-CD39-treated mouse MLN lymphocytes (Figure 5B,C), but the production of IL-4 was unaffected (Figure 5A). However, the data of systemic immune responses showed that comparable levels of IL-4, IL-5, and IL-13 were detected in the supernatants of OVA-cultured splenocytes from vector control and rAAV-CD39-treated mice (Figure 5D–F). The three OVA-specific antibodies in serum, IgE, IgG1, and IgG2a all showed comparable levels in the OVA-sensitized group (Appendix A). These results suggest that rAAV-CD39 treatment predominantly affects local rather than systemic immune responses.

### 3.5. Upregulated Percentage of Treg Cells in Draining Lymph Nodes of rAAV-CD39 Virus-Treated Mice

Given that CD39 functions by hydrolysis of ATP to AMP, this may bias the microenvironment toward immunosuppression, which then leads to the differentiation of regulatory T (Treg) cells [22,23]. Lymphocytes in MLNs were stained with fluorescent antibodies to CD3, CD4, CD25, and FoxP3, and then analyzed by flow cytometry. Increased percentage of Tregs was detected in MLNs of rAAV-CD39-treated mice (Figure 6A,C). This upregulation of Tregs might be caused by enhanced prevalence of indoleamine dioxygenase (IDO)^+^ plasmacytoid dendritic cells (pDCs, as PDCA-1^+^IDO^+^ cells, Figure 6B,D). These results suggested that rAAV-CD39 treatment had the capability to alleviate inflammatory response, possibly due to the increased Tregs in draining lymph nodes.

## 4. Discussion

To maintain host pulmonary tissue integrity, immune systems evolved not only a gas pedal but also a brake in responding to pathogen invasion or autoimmune disease. However, when an inappropriate response launches, such as an allergic response to harmless antigens, it breaks down the balance and even leads to a disease condition. For instance, the repetitive inflammatory responses to a danger signal impair the regulatory function and reinforce the effector function. In this study, we established an approach to reduce the level of danger signal extracellular ATP (eATP) and suppressed the allergic features in OVA-induced asthmatic mice. Our data demonstrated that CD39 gene transfer with recombinant adeno-associated virus (rAAV) vector system alleviated the asthmatic airway inflammation in OVA-sensitized mice. The expression of CD39 was rescued with rAAV-CD39 treatment to a comparable level of healthy controls. The most critical feature of the disease symptom, AHR, was significantly decreased. Notably, some local features of allergic inflammation, such as pulmonary eosinophilia, Th2 cytokines in BALF, and Th2 cytokines produced by OVA-stimulated MLN cells, were affected. However, systemic responses, including OVA-specific antibodies in serum and Th2 cytokines produced by splenocytes, were not affected with rAAV-CD39 treatment. These data suggest that rAAV vector-mediated CD39 expression can be a potential therapeutic approach for asthma.

DAMPs have been reported which are associated with several diseases and also have been pointed out to be potential targets for therapies. For instance, ATP is considered as a promising adjuvant in the treatment of nasopharyngeal carcinoma (NPC) [24] and HMGB1 is described as a target in some inflammatory diseases [25,26]. To restrict their proinflammatory ability, the extracellular DAMPs can be inactivated by the treatment of enzymes or neutralizing antibodies [27,28]. In line with these studies, our results share the same opinion to approve the idea that the eATP hydrolyzing enzyme is a potential target for asthma treatment.

The impact of high level eATP on the recruitment and activation of immune cells has been described in many inflammatory diseases, including chronic obstructive pulmonary disease (COPD), idiopathic pulmonary fibrosis, and asthma [8,9]. The release of eATP in bleomycin-treated MLE12 or BEAS-2B cells was also described [8]. The eATP levels are tightly regulated by ectonucleoside triphosphate diphosphohydrolases (NTPDases) and ecto 5′-nucleotidase (ecto 5′-NT; CD73) [10]. With the relatively lowest Km (10–17 µM) enzyme activity among all NTPDases [10], CD39 (NTPDase1) is suggested to be the most potent NTPDase. Indeed, the converse relationship between CD39 and asthmatic airway inflammation was firstly reported by Idzko et al. in 2007 [6]. The authors suggested that CD39 was a very critical factor in controlling the pulmonary eATP at low levels. In agreed with a study indicating that CD39 mRNA was decreased in PBMCs of patients with allergic asthma [14], our data found significant reduction of CD39 and accumulation of eATP in the lungs of asthmatic mice. Thus, inhibiting eATP concentrations in the allergen-induced lungs has the potential to relieve the inflammatory responses and asthma severity. Here, rescue of eATP hydrolyzing enzyme CD39 exhibits advantages of being easy to be processed and an increase of the opposite function by its metabolite. By upregulating the eATP hydrolyzing enzyme, the proinflammatory DAMP eATP was decreased. All these effects were mainly restricted in the target tissue. These properties anable the eATP-CD39 axis to become a potential condition for gene therapy against asthma.

Severe airway inflammation is usually the consequence of pulmonary leukocyte infiltration. The eATP levels contribute to the leukocyte infiltration and the knockout of CD39 induced more polymorphonuclear leukocyte (PMN) infiltration [29]. Some studies indicated that purinergic receptor P2Y_2_R participated in the accumulation of neutrophils, dendritic cells (DCs), and eosinophils [30,31]. Reduced eosinophilic infiltration was also reported in asthmatic mice with apyrase treatment [6,32]. The reduced airway inflammation of OVA-induced allergic C57BL/6 mice was achieved with the administration of exogenous apyrase prior to each OVA challenge [32]. These findings suggest that eATP and purinergic signaling played an important role in leukocyte infiltration in the lungs of asthmatic mice. Our data agreed with this perspective that the numbers of infiltrated eosinophils were modulated with the higher expression of CD39 and the lower eATP levels in rAAV-CD39-treated mice. However, two main chemokines for eosinophils, CCL11 and CCL24, were not affected. Whether the gene expression of CCL11 or CCL24 is regulated by eATP requires future studies. Nevertheless, our results are consistent with the findings of that the comparable level of CCL11, but reduced eosinophilia, was detected with lower eATP levels [6]. The lower expression of VCAM-1 with rAAV-CD39 treatment may contribute to the significant reduction of leukocyte infiltration in our asthmatic model mice. Taken together, rescued expression of CD39 by exogenous gene transduction was able to alleviate asthmatic airway inflammation via reduction of leukocyte infiltration.

With the reduction of ATP levels in BALF, we also noticed the asymmetric pattern of local Th2-type response (cytokines from OVA-activated MLN cell cultures) vs. systemic response (OVA-activated splenocyte cultures and serum antibodies). ATP was reported to promote inflammation and inhibit the generation of regulatory T cells (Tregs) [33]. In addition, adenosine was demonstrated to be a strong immunosuppressive molecule which is able to dampen the activation of T cell or induce the formation of Tregs [22,34]. Thus, the reduced ATP levels in the microenvironment may hint the potential for higher Treg cell activity. Furthermore, studies demonstrated that plasmacytoid DCs (pDCs) played a tolerogenic role in the asthmatic experimental model [35]. The expression of indoleamine 2,3-dioxygenase (IDO), which is a regulatory molecule leading the conversion of Tregs, can also be induced by adenosine receptor signaling [23,36]. These previous studies led us to test whether the microenvironment changes by rAAV-CD39 might be favored for Treg formation. We examined the concentrations of ATP and adenosine by LC-MRM, Tregs and DCs in MLNs. The results showed significant increased frequency of CD25^+^FoxP3^+^ T cells and higher, but not significant, pDCs in MLNs. These findings indicate that rAAV-CD39 treatment may not only affect the migration of leukocytes but also has the potential to strengthen the regulatory function locally.

The application of rAAV as efficient transfer systems had been more accepted in clinical and preclinical studies. Compared to other viral vectors, rAAV exhibits a great potential for gene therapy due to its safety, long-term expression capability without chromosomal integration and low immunogenicity [37,38]. One rAAV-based vector, LUXTURNA^TM^ (Voretigene neparvovec-rzyl; Spark Therapeutics, Inc.), has been approved by USA FDA for the treatment of Leber congenital amaurosis (LCA). There are also several ongoing phase 3/4 clinical trials using rAAV as the vector, including a study of long-term follow-up safety and efficacy study of spinal muscular atrophy Type 1, Type 2, or Type 3 (ClinicalTrials.gov: NCT04042025), a study to evaluate the efficacy and safety of a single sub-retinal injection of AAV2-REP1 in subjects with Choroideremia (ClinicalTrials.gov: NCT03496012), and a study to demonstrate the efficacy of AAV5-hFIXco-Padua (AMT- 061) and to further describe its safety profile (ClinicalTrials.gov: NCT03569891). We also established efficient and long-term expression systems to carry anti-inflammatory genes with the AAV9 cap protein, which specifically targets lung epithelial cells [19,20]. In this study, we attempted to target danger signal to control eATP by transferring CD39, without the need for repetitive injection of exogenous apyrase prior each OVA challenge [32]. Although other approaches are also able to lower eATP levels, such as the blockers of Panx-1 channel by carbenoxolone (CBX) or mimetic peptide ^10^panx1 [8,39,40], we believe that rAAV-CD39 vector can be an alternative treatment for asthma patients, having the advantages of stable and long-term gene expression, specific tissue tropism, and an impact on local immune responses.

In conclusion, a single treatment of rAAV-CD39 viral vector alleviated asthmatic airway inflammation on an OVA-sensitized mice model. The features of local response (cytokine in BALF and MLN cell culture) exhibited an asymmetric deduction compared to that of systemic response, suggesting it is a local modulator. As the first attempt in targeting proinflammatory DAMP eATP via gene transfer, this study supports the idea that CD39 is a potential candidate for the AAV-based gene therapy for asthma.

## 5. Conclusions

For the first time, the efficacy of rAAV vector targeting danger signal of the innate immune response to improve asthma was demonstrated. The hybrid rAAV2/9 vector viruses used in this study was composed with a cap protein that has tissue tropism to lung tissues and would limit the undesired systemic gene transfer. In addition, increased expression of membrane-bound apyrase CD39 on epithelial cells was able to alleviate inflammatory responses in OVA-induced asthmatic mice. This treatment reduced the inflammation by reducing the purinergic signaling-ATP, Th2-type cytokines, and the increase of regulatory T cells. We believe that the rAAV vector mediated expression of CD39 may be a potential treatment for asthma.

## Figures and Tables

**Figure 1 biomedicines-09-00656-f001:**
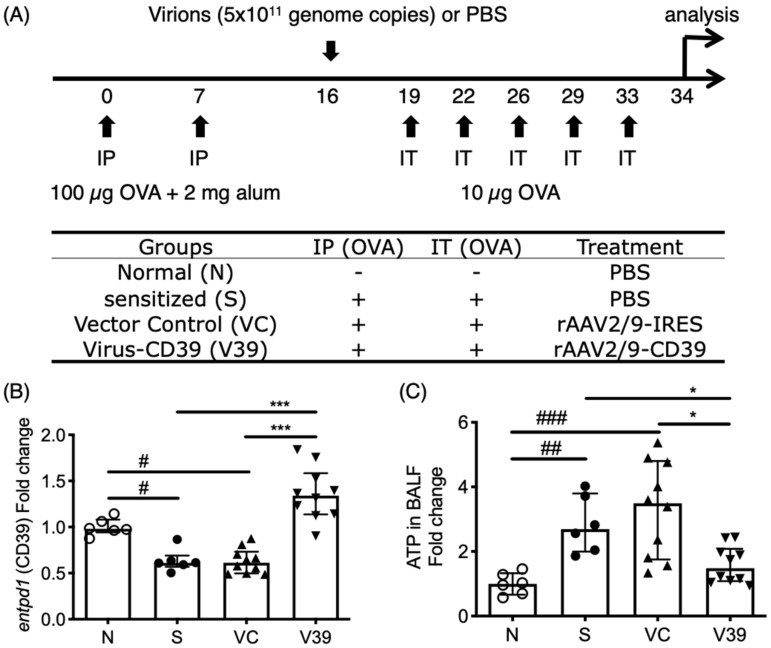
ATP levels in bronchoalveolar lavage fluid (BALF) of rAAV-CD39 treated mice was downregulated with rescued expression of CD39. (**A**) Brief protocol for OVA sensitization and rAAV-CD39 treatment. Mice were sensitized by intraperitoneal injection (IP) of OVA/alum on days 0 and 7 and were challenged by OVA 5 times within 2 weeks. rAAV-CD39 was intratracheally (IT) administrated on day 16. (**B**) The relative expression level of CD39 (gene symbol: *entpd1*) to β-actin (*actb*) in the lungs was increased in the rAAV-CD39 treated group, compared to the controls. (**C**) The relative ATP level to the normal controls in BALF was decreased in rAAV-CD39 treated group. ATP levels were determined with ATPlite analysis. Data are presented as median with interquartile range (* *p* < 0.05, *** *p* < 0.001 in the comparison of V39 with S or VC; # *p* < 0.05, ## *p* < 0.01, ### *p* < 0.001 in the comparison of N with S or VC).

**Figure 2 biomedicines-09-00656-f002:**
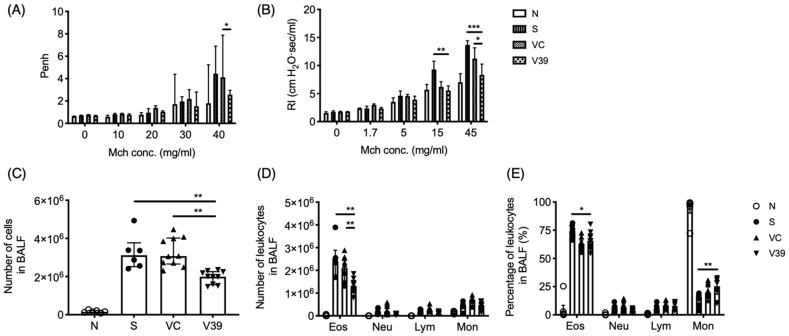
Exogenous expression of CD39 suppressed asthmatic airway inflammation. After the last OVA exposure, several cardinal features were analyzed. Airway hyperresponsiveness were determined by (**A**) Whole Body Plethysmography and (**B**) Resistance and Compliance (the difference was calculated by two-way ANOVA followed by Bonferroni test). Leukocyte infiltration in BALF was determined in (**C**) total cell numbers and differential leukocyte populations were performed by (**D**) the cell numbers and (**E**) the proportions. Data are presented as median with interquartile range (4–8 mice for Penh; 9–12 mice for RI; * *p* < 0.05, ** *p* < 0.01, *** *p* < 0.001 in the comparison of V39 with S or VC).

**Figure 3 biomedicines-09-00656-f003:**
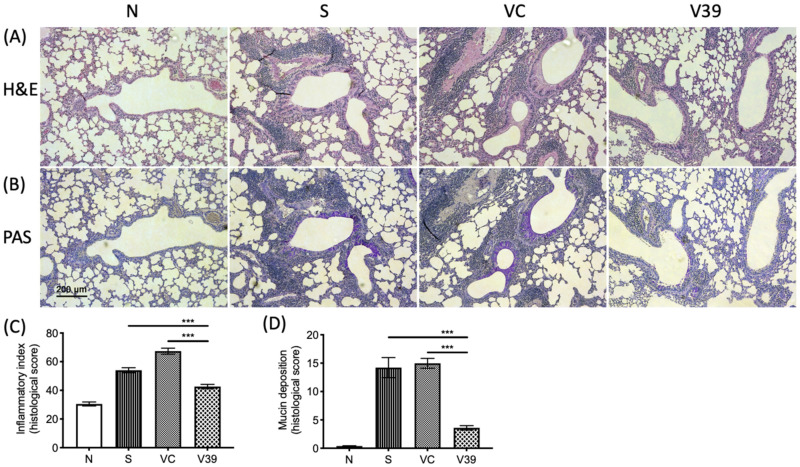
Inflammatory cell infiltration and mucin deposition was decreased by the treatment of rAAV-CD39. The Formaldehyde-fixed, paraffin-embedded lung samples were sliced and stained by (**A**) hematoxylin and eosin (H&E) and (**B**) periodic acid-Schiff (PAS). The quantitative data of each slide was analyzed by ImageJ and presented as a histological score, including (**C**) inflammatory index and (**D**) mucine deposition (the purple-magenta color in B). There were 5 areas in 2 fields from 4–6 slides applied for quantification. Scale bar = 50 µm. Data are presented as a median with an interquartile range (*** *p* < 0.001 in the comparison of V39 with S or VC).

**Figure 4 biomedicines-09-00656-f004:**
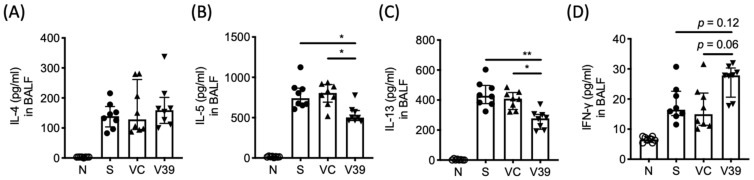
The levels of Th2 cytokines were downregulated in BALF with exogenous expression of CD39. After the last OVA exposure, the lungs were lavaged and the concentration of (**A**) IL-4, (**B**) IL-5, (**C**) IL-13, and (**D**) IFN-γ in BALF were determined by ProcartaPlex. Data are presented as median with interquartile range (* *p* < 0.05, ** *p* < 0.01 in the comparison of V39 with S or VC).

**Figure 5 biomedicines-09-00656-f005:**
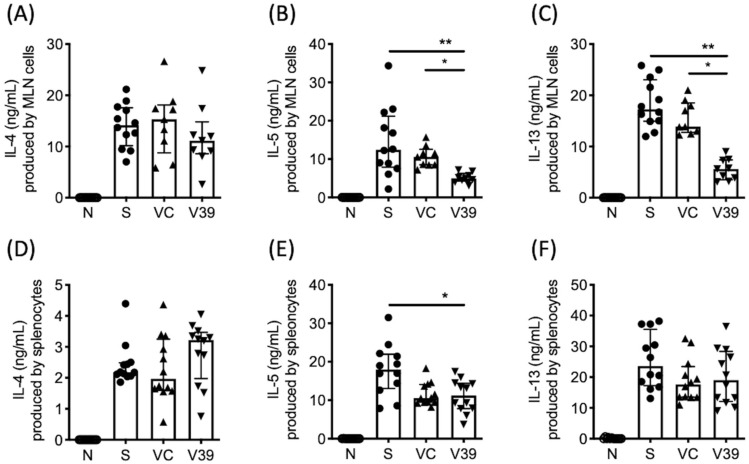
Cytokine levels in the culture supernatants of OVA-stimulated cells of mediastinal lymph nodes (MLNs) and spleens. Lymphocytes from (**A**–**C**) MLN or (**D**–**F**) spleen were stimulated with 100 µg/mL OVA for 6 days. Cytokine levels were measured by ELISA. Data are presented as median with interquartile range (* *p* < 0.05, ** *p* < 0.01 in the comparison of V39 with S or VC).

**Figure 6 biomedicines-09-00656-f006:**
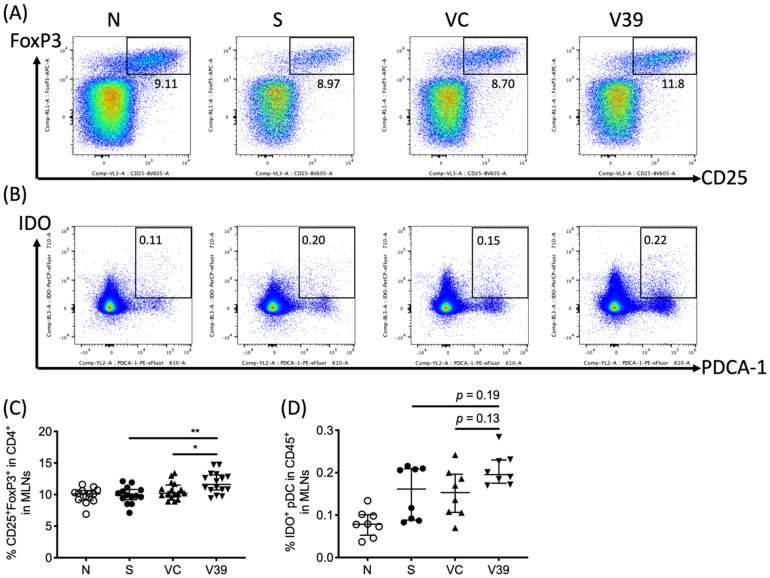
The frequency of CD25^+^FoxP3^+^ CD4 regulatory T cells and tolerogenic dendritic cells were increased in draining lymph nodes. Leukocytes in mediastinal lymph nodes were suspended, and the representative flow plots of (**A**) CD25^+^FoxP3^+^ cells in CD4 T cells and (**B**) indoleamine 2,3-dioxygenase (IDO) expressing plasmacytoid dendritic cells (pDC; PDCA-1^+^) in leukocytes were determined by flow cytometry. The quantitative data of (**C**) Tregs and (**D**) IDO^+^ pDCs are shown in percentages. Data are presented as median with interquartile range (* *p* < 0.05, ** *p* < 0.01 in the comparison of V39 with S or VC).

## Data Availability

All data are contained in this article.

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
