# Peer review of "Reducing Lung ATP Levels and Alleviating Asthmatic Airway Inflammation through Adeno-Associated Viral Vector-Mediated CD39 Expression"

_biomedicines, 2021, doi:10.3390/biomedicines9060656_

Round 1
Reviewer 1 Report
In this study, the authors demonstrated that recombinant adeno-associated virus (rAAV)-mediated CD39 gene transfer into OVA-mediated asthmatic mice effectively reduced airway inflammation by reducing the purinergic signaling-ATP, type 2 cytokines, as well as by increasing regulatory T cells. Overall, the authors have made a good attempt at challenging the possibility of an entirely novel asthma treatment through gene therapy. The manuscript is well-written, and the experiments are well-designed and conducted. In particular, the value of this study may be enhanced by that the authors have confirmed that this therapy has only a local effect, assuming its practical application as gene therapy.
I have some minor comments as described below.
1) Figure 1(A): “GC” at the top should be spelled out as “Genome copies”.
2) Line 174: It should be described as "intratracheally (IT) administrated" like "intraperitoneal (IP) injection" at line 172.
3) The statistical results in Figures and Figure legends are not clear.
For example, in Figure 4C, there is an asterisk between "VC vs. V39", i.e., P < 0.05. However, in Figure 4B, between "VC vs. V39", there is a line but no *. Does this mean that there was “no significance”, or is it just a case of forgetting to fill in the *?
The same problem can be seen in all figures except Figure 5 and Figure 6C and D. Please correct them.
4) Line 185: The rAAAV-CD39-treated => The rAAV-CD39-treated
5) Lines 304-305: I think the most important technical aspect of this study is that the expression of CD39 has been restored to a comparable level of healthy controls, not overexpression. In the experiments of this study, the authors analyzed two and a half weeks after the gene transfer. Do you have any information about the kinetics of the introduced CD39 expression levels during this period?
Author Response
1) Figure 1(A): “GC” at the top should be spelled out as “Genome copies”.
Reply: We appreciate the reviewer’s comment, and we have changed the illustration of GC into genome copies in Figure 1.
2) Line 174: It should be described as "intratracheally (IT) administrated" like "intraperitoneal (IP) injection" at line 172.
Reply: We agreed that it should be described in this way. We have changed the description at line 174.
3) The statistical results in Figures and Figure legends are not clear.
For example, in Figure 4C, there is an asterisk between "VC vs. V39", i.e., P < 0.05. However, in Figure 4B, between "VC vs. V39", there is a line but no *. Does this mean that there was “no significance”, or is it just a case of forgetting to fill in the *?
The same problem can be seen in all figures except Figure 5 and Figure 6C and D. Please correct them.
Reply: We appreciate the reviewer’s concern. In that case, the comparison of S vs. V39 and VC vs. V39 share the same statistical level with one asterisk sign. That was the reason why we only labeled on the top line.
We agreed the point raised by the reviewer, to label the sign of significance on each pair to make the data much more clear. Therefore, we revised the labels in all of the corresponding figures.
4) Line 185: The rAAAV-CD39-treated => The rAAV-CD39-treated
Reply: We appreciate the reviewer’s reminder and we had corrected this typing mistake.
5) Lines 304-305: I think the most important technical aspect of this study is that the expression of CD39 has been restored to a comparable level of healthy controls, not overexpression. In the experiments of this study, the authors analyzed two and a half weeks after the gene transfer. Do you have any information about the kinetics of the introduced CD39 expression levels during this period?
Reply: We appreciate the reviewer’s concern. Indeed, our data did not support us to use the word over-expression. With the concern of CD39 expression in treated group only reaching the comparable level of healthy control, we believe it is caused by the cell types that can be infected by rAAV2/9. At the beginning of the development of AAV vector in our lab, we had tested the location of expressed hrGFP reporter by IHC. The data revealed that hrGFP+ cells were mainly lung epithelial cells. Although CMV promoter is a powerful promoter to induce a strong expression of exogeneous genes in transduced cells, other types of cells without the transduced gene masked the signal of exogenous gene expression in the lung tissues. Regarding to the concern of the kinetic of the rAAV2/9 vector, we also had tested the expression of the hrGFP reporter on different time points. The results indicated that the reporter expression reached the maximal level three days after the administration of AAV vector and maintained the expression up to at least three months. This is the main reason for us to perform the gene therapy three days before the first OVA challenge. The information was reported in our previous work, Hum Gene Ther. 2012 Nov;23(11):1156-65.
Reviewer 2 Report
The authors present an interesting manuscript about “Reducing lung ATP levels and alleviating asthmatic airway inflammation through adeno-associated viral vector-mediated CD39 expression “. The results show that the rescued CD39 expression in mice with administration of rAAV-CD39 160 and the ATP levels in bronchoalveolar lavage fluid were decreased. The manuscript is well written and the figures are balanced. The manuscript is sound. I suggest accepting it in the present form. I suggest adding data on the number of mice enrolled and on the median age
Author Response
We appreciate the reviewer’s comments. We have shown the number of mice in each experiment as either the dots in each bar of Figures 1, 2C-E, 4, 5, and 6 (each dot represents one mouse) or being described as n numbers in the legends to Figures 2A-B, and 3. The age of mice enrolled in the experiments was 8-10 weeks and this information was described in the first paragraph of materials and methods.